# The beetle amnion and serosa functionally interact as apposed epithelia

Maarten Hilbrant, Thorsten Horn, Stefan Koelzer, Kristen A Panfilio*

Institute for Developmental Biology, University of Cologne, Cologne, Germany

**Abstract** Unlike passive rupture of the human chorioamnion at birth, the insect extraembryonic (EE) tissues – the amnion and serosa – actively rupture and withdraw in late embryogenesis. Withdrawal is essential for development and has been a morphogenetic puzzle. Here, we use new fluorescent transgenic lines in the beetle *Tribolium castaneum* to show that the EE tissues dynamically form a basal-basal epithelial bilayer, contradicting the previous hypothesis of EE intercalation. We find that the EE tissues repeatedly detach and reattach throughout development and have distinct roles. Quantitative live imaging analyses show that the amnion initiates EE rupture in a specialized anterior-ventral cap. RNAi phenotypes demonstrate that the serosa contracts autonomously. Thus, apposition in a bilayer enables the amnion as 'initiator' to coordinate with the serosa as 'driver' to achieve withdrawal. This EE strategy may reflect evolutionary changes within the holometabolous insects and serves as a model to study interactions between developing epithelia.

## Introduction

Embryogenesis requires dynamic interaction between tissues to create changing three-dimensional configurations, culminating in the completion of the body. In parallel to the amniote vertebrates (*Calvin and Oyen, 2007*), the insects have evolved extraembryonic (EE) tissues that arise in early embryogenesis to envelop the embryo (*Panfilio, 2008*). These are the amnion and the serosa, which are both simple, squamous epithelia. Like its vertebrate namesake, in most insect species the amnion encloses a fluid-filled cavity around the embryo. As the outermost cellular layer, the serosa provides mechanical and physiological protection (*Farnesi et al., 2015*; *Jacobs et al., 2013*; *2014*; *Rezende et al., 2008*). This protective configuration is not permanent, though, and a major reorganization of the EE tissues is essential for embryos to correctly close their backs in late development.

Reorganization involves whole tissue eversion, contraction, and final apoptosis of both EE tissues (*Panfilio et al., 2013*). For these events, perhaps the nearest morphogenetic equivalent in the model system *Drosophila* is eversion of the wing imaginal disc during metamorphosis, where the squamous peripodial epithelium also exhibits these behaviors (*Aldaz et al., 2010*). However, research on *Drosophila* cannot address the morphogenesis of the two EE epithelia directly, due to the secondarily derived nature of its single EE tissue, the amnioserosa, which does not surround the embryo (*Rafiqi et al., 2012*; *Schmidt-Ott, 2000*).

Insect EE withdrawal – the active process whereby the EE tissues withdraw from the embryo and leave it uncovered – has been addressed at the level of gross morphology in classical descriptions for many species (reviewed in *Panfilio, 2008*). However, a major open question has been the organization and role of the amnion. This is primarily because it is difficult to visualize in its native topography with respect to other tissues. A lack of amnion-specific molecular markers (discussed in *Koelzer et al., 2014*) and the histological similarity and proximity of the serosa (*Panfilio and Roth, 2010*; *van der Zee et al., 2005*) have been particular challenges.

*For correspondence: kristen. panfilio@alum.swarthmore.edu

Competing interests: The authors declare that no competing interests exist.

**eLife digest** Early in development a protective fluid-filled sac forms around an embryo. In humans, this sac bursts during birth, but the sac surrounding insect embryos ruptures long before these animals begin to emerge from their eggs. This early rupture is important for insects to develop normally: if an insect embryo's sac remains intact too long, the animal's back will not close properly.

The sac that surrounds insect embryos has two layers: an inner layer called the amnion, and a tough outer layer called the serosa. However, it has been difficult to study what happens to the amnion as the insect embryo develops because it is hard to distinguish it from the serosa. Now, Hilbrant et al. have used genetically engineered red flour beetles in which the cells of the amnion produce a fluorescent protein that can be viewed under a microscope. This allowed the amnion to be observed in living specimens during beetle embryo development, and revealed that the amnion attaches and detaches from the serosa more than once as the embryo develops. Furthermore, the amnion and serosa remain as distinct tissues as they withdraw from the embryo.

Hilbrant et al. also found that the cells in part of the amnion near the head of the beetle embryo have a special shape before the sac ruptures. This region of the amnion breaks apart first, and the serosa breaks open a few minutes afterwards. Once the two layers have broken, they pull back from the embryo like a pillowcase being turned inside out as it is removed from a pillow. In normal beetles, this process is quite rapid and squeezes the embryo's abdomen. But in beetles genetically manipulated to lack a serosa the process is slower because the amnion is not strong enough by itself to squeeze the embryo.

Overall, the experiments show that the amnion starts the rupture of the red flour beetle embryo's protective sac and the serosa drives the process of the sac being peeled back. Further research will now investigate the mechanics behind the two tissues' roles and whether the amnion and serosa display similar behaviors in other related insect species.

Here, we present the first clear determination of the relative topography and role of the amnion in late development in a holometabolous insect, the red flour beetle, *Tribolium castaneum*. We characterize an enhancer trap line that labels the amnion and use this in conjunction with recently characterized serosal lines (*Koelzer et al., 2014*) to morphogenetically dissect which tissue is responsible for which aspects of EE tissue withdrawal. The topographical arrangement of the tissues differs strikingly from what was previously known in hemimetabolous insects and what had previously been hypothesized for *Tribolium*. To better appreciate the implications of this arrangement for morphogenesis, we situate these observations in the larger context of EE development at preceding and following stages. This global, mesoscopic approach to evaluating tissue interactions significantly improves our understanding the entire withdrawal process, including the first detailed examination of EE rupture in any insect. Furthermore, we provide evidence that while the serosa strongly drives the contraction and folding of the tissues, the amnion initiates EE rupture.

## Results and discussion

### The amnion and serosa form a bilayer during late development

To augment the toolkit for tissue-specific visualization in *Tribolium*, we identified and characterized an enhancer trap line with amniotic EGFP expression (*Figure 1*, see also *Figure 1—figure supplement 1*, *Video 1*). Prior to withdrawal morphogenesis, the EGFP-labeled tissue fully envelops the embryo but does not cover the yolk (*Figure 1A–B*). To confirm that this tissue is indeed the amnion, and not a specialized region of the serosa, we examined EGFP expression after RNAi for *Tc-zen1*, thereby eliminating serosal tissue identity (*van der Zee et al., 2005*). In the absence of the serosa, the amnion occupies a dorsal position over the yolk, and indeed this tissue expresses EGFP (*Figure 1C–D*).

We then used the tissue-specific EE imaging lines to address the arrangement of the amnion and serosa during late development. Initially the two EE tissues are physically separate (*Handel et al.,*

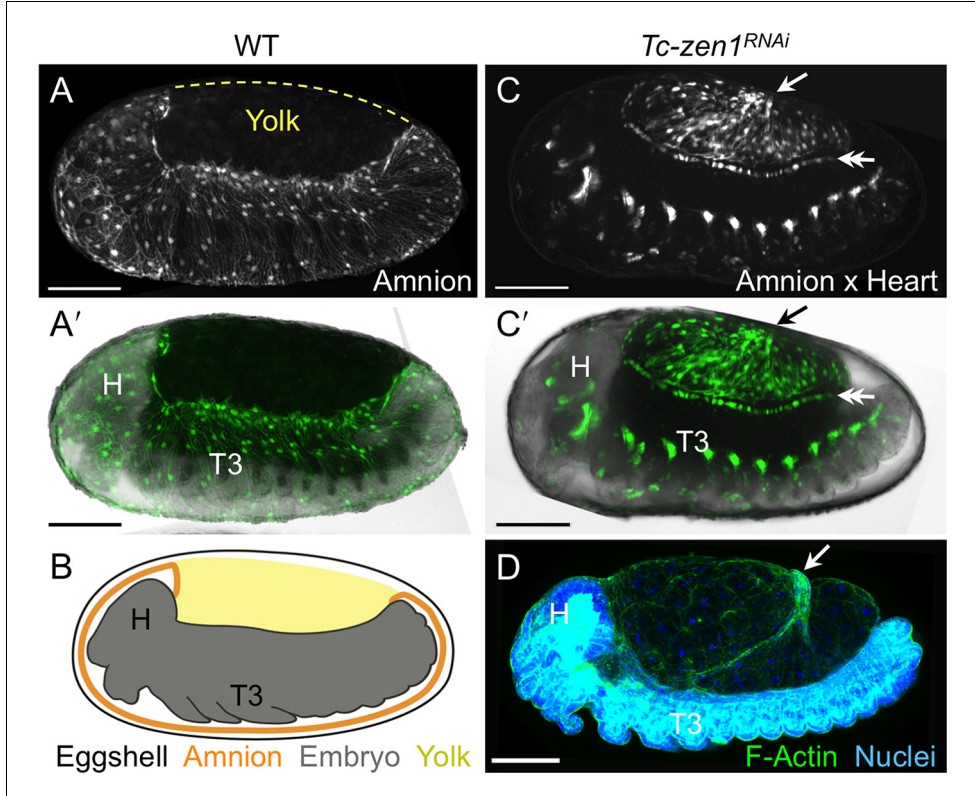

**Figure 1.** The enhancer trap line HC079 is an autonomous amniotic tissue marker. Images are lateral, with anterior left and dorsal up, shown as maximum intensity projections or a mid-sagittal schematic. Visualization reagents are indicated. (**A–B**) In wild type (WT), EGFP expression is extraembryonic (EE) in a ventral domain that fully covers the embryo but not the yolk prior to rupture. See also *Figure 1—figure supplement 1*, *Video 1*. (**C–D**) Consistent with the WT EGFP domain being amniotic, the entire EE tissue expresses EGFP when serosal identity is eliminated after *Tc-zen1^RNAi^*. Here, the EE tissue does cover the yolk (dorsal to the cardioblast cell row: double-headed arrow), and, during the mid-withdrawal stage shown here, acquires a diagnostic 'crease' (arrow) (*Panfilio et al., 2013*). Scale bars are 100 μm. Abbreviations: H, head; T3, third thoracic segment.

The following figure supplement is available for figure 1:

**Figure supplement 1.** Developmental time course of EGFP expression in the enhancer trap line HC079.

*2005*), but they progressively come together during early germband retraction until the only clearly discernible amniotic region is a rim of tissue at the embryo's dorsal margin (*Figure 2A-A2*). Previous histological studies in *Tribolium* and other holometabolous insects concluded that the region of overlap throughout the ventral half of the egg comprised a single EE cell layer, the EE tissues having intercalated or otherwise 'fused' (e.g., *Kobayashi and Ando, 1990*; *Patten, 1884*; *van der Zee et al., 2005*). Alternatively, this structure was interpreted as only serosa (*Panfilio et al., 2013*), as the underlying amniotic region undergoes apoptosis in a hemimetabolous insect, the milkweed bug *Oncopeltus fasciatus* (*Panfilio and Roth, 2010*). In fact, we find that both tissues persist as apposed and very thin but distinct squamous epithelial layers that are evident in both sectioned and whole mount material (*Figure 2B–E*, *Figure 2—figure supplement 1*). Apposition occurs over the entire surface area of the amnion except its dorsal rim, such that the serosa and amnion form an extensive epithelial bilayer. Several hours after the completion of germband retraction, the EE membranes rupture and withdraw from the embryo (*Koelzer et al., 2014*). Interestingly, the amnion-serosa bilayer is maintained throughout withdrawal morphogenesis, with all major folds and minor bends involving both EE tissue sheets (*Figure 3*). The new amnion EGFP line thus sheds light on EE tissue structure during withdrawal: rather than a single, pseudostratified tissue, each EE tissue persists as a monolayer, representing a novel EE tissue organization compared to what is known for other

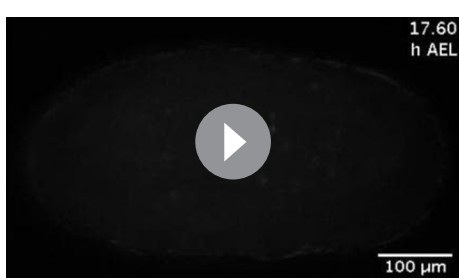

**Video 1.** Time course of EGFP expression in the *Tribolium* enhancer trap line HC079. The embryo is shown in lateral aspect with anterior left and dorsal up. Strengthening EGFP signal is specific to the amnion during germband retraction and up to the point of EE tissue rupture and withdrawal morphogenesis. In the later stages, additional embryonic expression domains occur in the eyes and legs. Throughout the movie, wandering yolk globules also exhibit low levels of EGFP, a feature observed for other enhancer trap lines from this screen (*Koelzer et al., 2014*). The time stamp specifies minimum age from a four-hour egg collection, in hours after egg lay (h AEL). Maximum intensity projections from a deconvolved z-stack (5 µm step size for a 60 µm stack) recorded every ten minutes over a 48-hr time-lapse at 30°C, acquired with an inverted DeltaVision RT microscope (Applied Precision). Scale bar is 100 µm. Selected still images are shown and further described in *Figure 1—figure supplement 1*.

species. Given that the amnion persists as a discrete tissue throughout withdrawal morphogenesis, we then investigated its role in this process, starting with its structure at the onset of rupture.

## The cellular structure and dynamics of rupture

While serosal cells overlying the amnion are fairly uniform in size and shape (*Figure 4A-A2*), close inspection of the amnion EGFP signal shows that an anterior-ventral cap of amnion cells is morphologically distinct (*Figure 4B*). These large cells are rounder and have sharper cell outlines compared to cells elsewhere in the tissue, which are striated and elongated along the dorsal-ventral axis. The anterior-ventral amniotic cells also have brighter EGFP signal (*Figure 1—figure supplement 1*). Moreover, we consistently observe that EE rupture begins within this territory (*Figure 4C*, see also *Video 2*). Definitive opening only directly affects a handful of amniotic cells, and the exact location of opening within the anterior-ventral cap varies from lateral (*Figure 4C*) to more medial and anterior sites (*e.g., Figure 2—figure supplement 1D–E*). This suggests that the entire anterior-ventral cap constitutes a rupture competence zone.

The amniotic cells at the opening exhibit impressive plasticity in both cell and nuclear shape on very short time scales as they are stretched in multiple directions. Remarkably, definitive rupture initiates not only by intercellular separation but in some cases by intracellular hole formation (*e.g., Figure 4D1-D2*: red cell). Among recordings with sufficient spatial and temporal resolution, we followed intracellular opening in three specimens. In the example shown here, thin cytoplasmic remnants of the affected cell that were distant from the nucleus and main cell body could be tracked initially, but these may fragment as the opening enlarges to the circumference of the egg. However, the nuclei (and main cell body) of cells at the site of rupture can be followed during withdrawal and appear to contribute to the persistent edge of the tissue (also in *Video 2*). These cellular contortions do not occur in the serosa, where cells at the site of rupture separate from their neighbors by spatial translation without substantial deviation from regular, polygonal cell shapes with centrally positioned nuclei (*Figure 4E1-E2*). Meanwhile, at no time prior to or during rupture did we observe cell loss in the amnion (see also *Figure 2—figure supplement 1E*). There is also no discernible change in apical area or EGFP brightness of amniotic cells directly involved in rupture compared to cells elsewhere in the amniotic cap, such that the onset of rupture appears very abrupt. Thus, rupture in *Tribolium* within a specialized competence zone of the amnion is fundamentally different to the strategy in *Oncopeltus*. In the latter species, gradual apoptosis of the amnion at the rupture site begins over a day (16% development) before rupture occurs but is in itself insufficient as a proximate trigger, which remains unknown (*Panfilio and Roth, 2010*).

## The EE tissues rupture at different rates

In addition to cell shape changes, we also quantified the rate of EE tissue rupture. We find that rupture initiation in *Tribolium* – defined as the short interval of initial opening of the EE membranes – is significantly longer in the amnion than in the serosa (*Figure 4F-G*, [p<0.001, Mann-Whitney U test; see Materials and methods for details of the landmarks that delimit this interval). This is also true even when slight differences in developmental rate between genetic backgrounds are considered

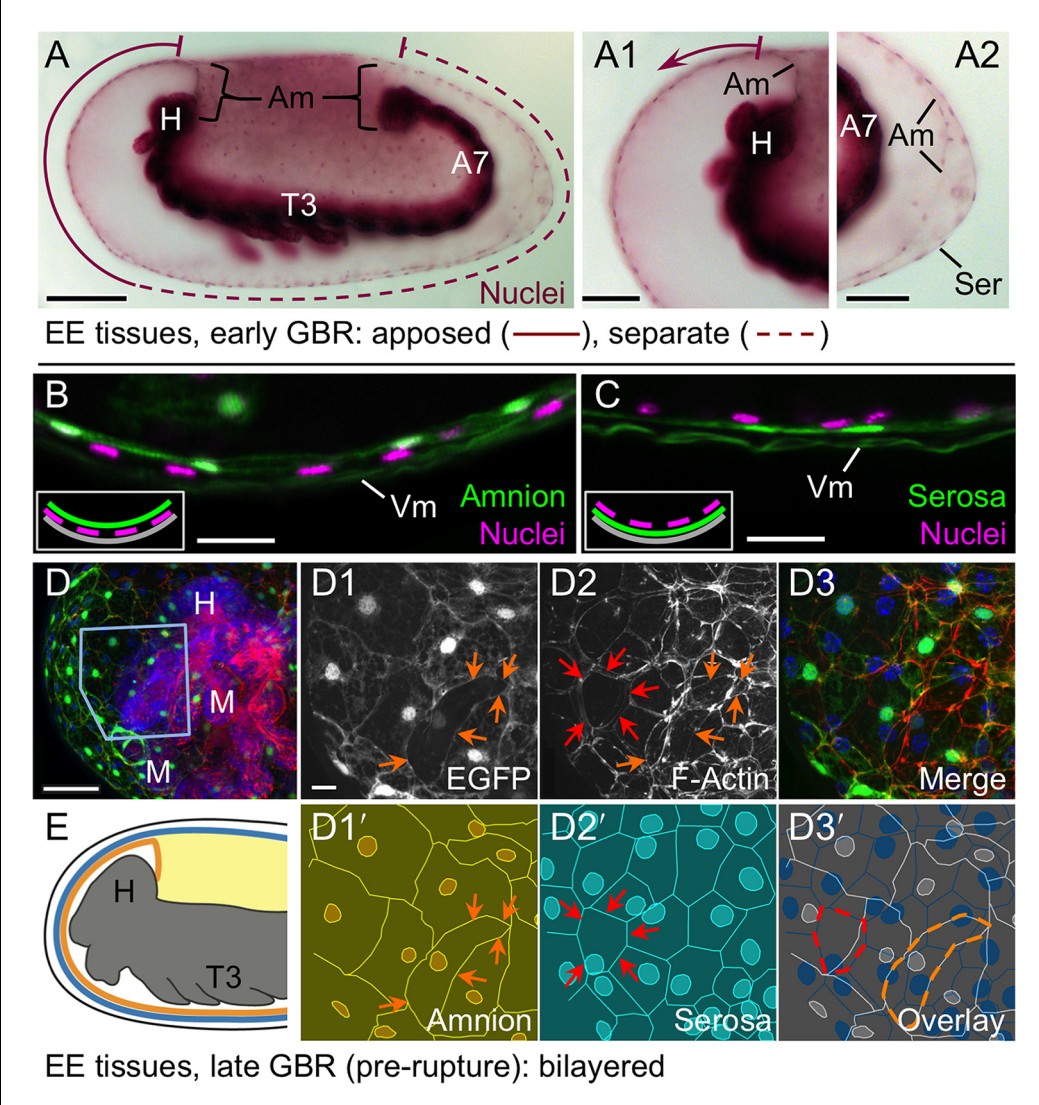

**Figure 2.** The amnion and serosa form a bilayer during the germband retraction stage. Images are lateral (**A–C,E**) or ventral-lateral (**D**), with anterior left and dorsal up. (**A**) During germband retraction, the amnion progressively comes together with the serosa, shown sagittally at an intermediate stage when the tissues are apposed anteriorly (**A1**) but still separate posteriorly (**A2**). Fuchsin preparation causes embryo shrinkage (*Wigand et al., 1998*), amplifying apparent amniotic cavity volume, but without altering tissue topography, which is consistent across dozens of stage-matched specimens. (**B–D**) Before rupture, the ventral EE tissue under the eggshell (autofluorescent vitelline membrane) is comprised of distinct serosal (outer) and amniotic (inner) layers. In sagittal sections (**B,C**), tissue-specific EGFP labels continuous tissue sheets, while nuclei (DAPI stain) of the apposed EE tissue remain EGFP-negative and in a separate layer (inset schematic). Maximum intensity projections (**D**) also show two epithelial layers, which can be distinguished by tissue-specific cellular morphologies. The box in the first panel indicates the magnified region in **D1**-**D3**. Amnion-specific EGFP illuminates the nuclei and cell boundaries in this tissue (**D1,D1′**). A phalloidin counterstain shows a complex network of F-Actin filaments, including weak signal for amniotic cell boundaries (orange arrowheads), and a distinct pattern of thicker filaments in a double-walled, polygonal arrangement that corresponds to serosal cell boundaries (**D2,D2′**, see *Figure 2—figure supplement 1* for serosal EGFP labeled specimens and additional details). Finally, comparison of a nuclear counterstain (DAPI) with the EGFP nuclear signal distinguishes the nuclei of the two EE tissues (here, EGFP-negative nuclei are serosal, shown in **D2′**), providing the information for a schematic overlay of these distinct tissues (**D3,D3′**: outlined cells are the same as those indicated by arrows in the previous panels). (**E**) The EE bilayer is illustrated in mid-sagittal view of the anterior, according to the color scheme in *Figure 1B* and with the serosa shown in blue. Scale bars are 100

*Figure 2 continued on next page*

*Figure 2 continued*

µm (**A**), 50 µm (**A1-A2,D**), and 10 µm (**B,C,D1-D3**). Abbreviations: A7, seventh abdominal segment; Am, amnion; GBR, germband retraction; M, mandible; Ser, serosa; Vm, vitelline membrane; and as defined in *Figure 1*.

The following figure supplement is available for figure 2:

**Figure supplement 1.** The amnion and serosa form a persistent bilayer comprised of two morphologically distinct epithelia, including in the anterior-ventral rupture competence zone.

(see also *Figure 1—figure supplement 1*). This result, supported by the distinct morphological appearance of the amnion at the site of rupture (*Figure 4B*), suggests that the amnion initiates rupture. The serosa then ruptures after a slight delay of a few minutes. Indeed, the difference in initiation duration corresponds morphologically to only a slight opening of the amnion, generally no larger than the area of the anterior-ventral cap region (82%, *n* = 28, data set from *Figure 4F*), before the serosa would also perceptibly rupture and then catch up with the amnion (*Figure 4G*, *Figure 4—*

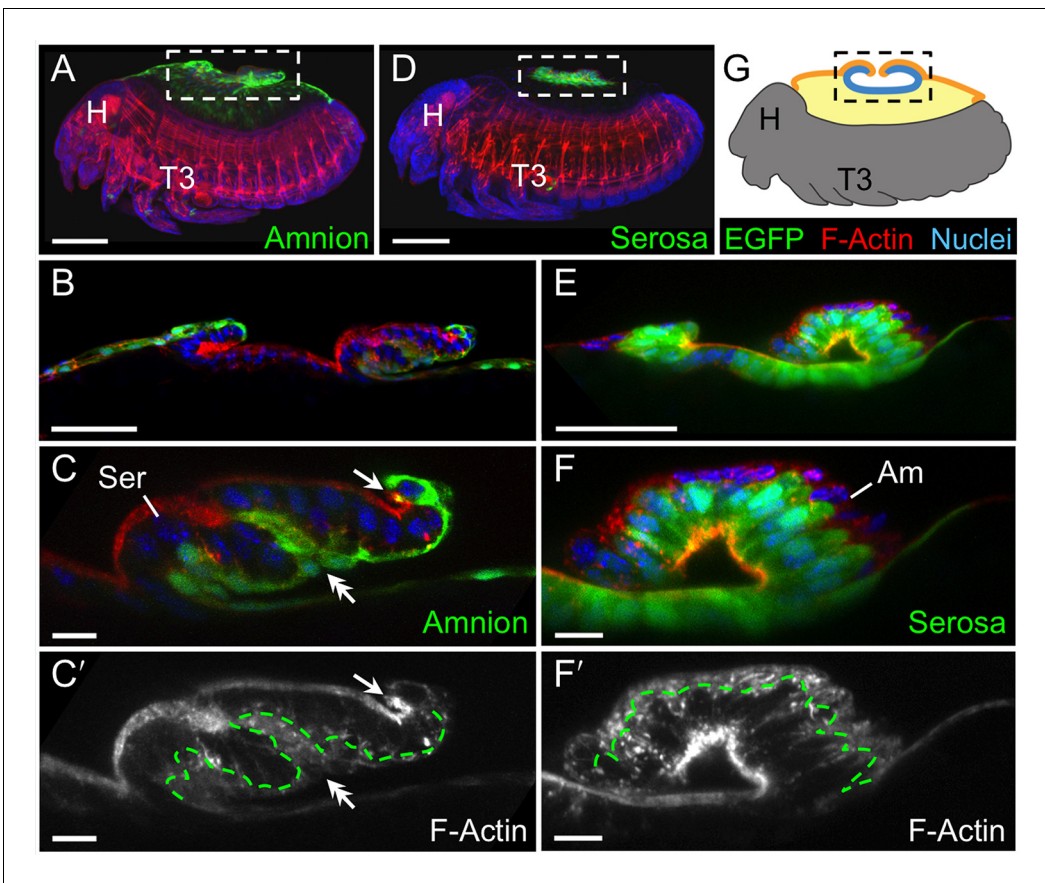

**Figure 3.** The extraembryonic bilayer moves as a single unit throughout late morphogenetic remodeling. Images are lateral, with anterior left and dorsal up, shown as maximum intensity projections (**A,D**), sagittal optical sections (**B,C,E,F**), or as a mid-sagittal schematic (**G**). (**A–G**) The bilayered EE structure is maintained during early serosal compaction. The embryos shown here are at a stage when the serosa folds medially, such that mid-sagittal sections show anterior and posterior arms to the folding tissue (**B,E,G**). Higher magnification images of cellular structure focus on the posterior arms (**C,F**). Panels **A–C** and **D–F** each show a single embryo. Annotations: arrow, ruptured edge of both tissues; double-headed arrow, double bend in the tissues; dashed green lines delimit EGFP domains. Scale bars are 100 µm (**A,D**), 50 µm (**B,E**), and 10 µm (**C,C′,F,F′**). Abbreviations and schematic color scheme as in previous figures.

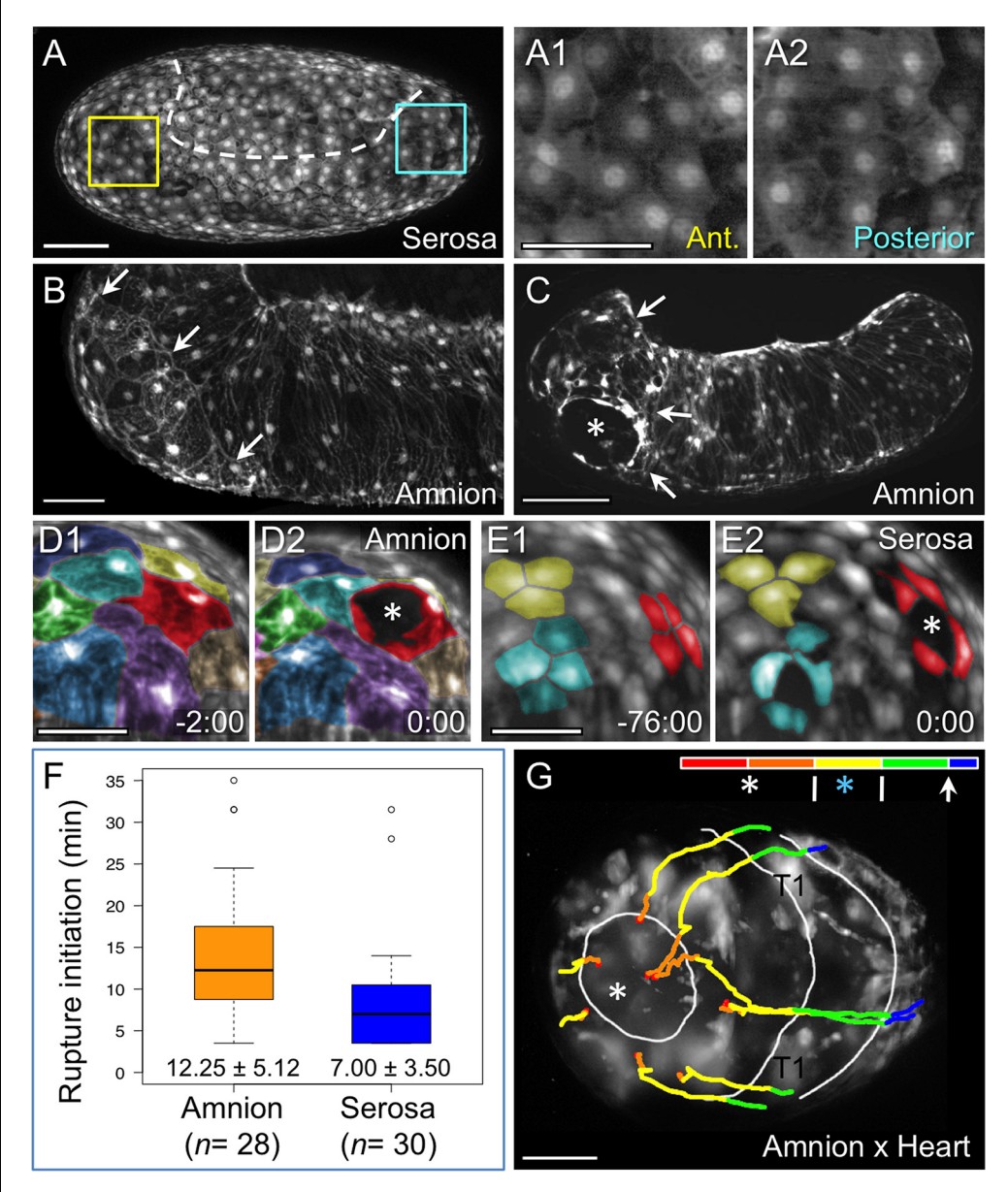

**Figure 4.** Rupture dynamics: amniotic initiation. Images are maximum intensity projections in lateral (**A–C**), anterior (**D–E**), or anterior-ventral (**G**) views. (**A**) The serosa maintains a homogeneous cellular morphology throughout the region apposed to the amnion (below the dashed white line, compare with *Figure 1A'*), shown in detail for anterior (Ant., **A1**) and posterior (**A2**) regions. (**B–C**) In contrast, amnion-EGFP signal before (**B**) and during (**C**) rupture highlights an anterior-ventral cap of morphologically distinct cells (arrows); the asterisk marks the tissue opening (see also *Video 2*). (**D–E**) Cell shape changes around the site of opening (asterisk, red cells) differ markedly between amniotic cell contortions (**D**) and more regular serosal cell shapes (**E**). Colors mark unique amniotic cells or groups of 3–4 neighboring serosal cells. The nature of opening and cellular connections is further described in the main text, and long-term cell tracking is shown in *Figure 2—figure supplement 1*. Time stamps are in minutes:seconds at 30°C, relative to rupture at 0:00. The entire interval from initial opening until the EE tissue fully cleared the head was 8 min (**D**) and 4 min (**E**). (**F**) Box plot showing that the rupture initiation interval, as defined in the Materials and methods section, is longer in the amnion (values are median ± median absolute deviation, at 27.5–28°C). (**G**) Tissue opening in a heterozygote embryo permits tracking of the ruptured EE tissue relative to embryonic anatomical landmarks ('T1' labels the proximal region of the first leg pair). The colored time scale shows duration of track segments over 36.7 min at 19.5 ± 1°C. Along the time scale and superimposed on the embryo are the site of rupture (white asterisk) and the withdrawing EE tissue edge (white lines, line with

*Figure 4 continued on next page*

*Figure 4 continued*

arrowhead marks time point shown). Tracks are shown for selected nuclei. For comparison, rupture is also indicated along the time scale for a morphologically stage matched embryo with serosal EGFP (blue asterisk; see also *Figure 4—figure supplement 1*, *Video 3*), which shows a shorter rupture initiation interval (interval from the asterisk to the right end of the colored time scale), consistent with the data in panel F. Scale bars are 100 µm (**A,C**) and 50 µm (**A1–A2,B,D1–D2,E1–E2,G**). Panel **B** shows the same embryo as in *Figure 1A*.

The following figure supplement is available for figure 4:

**Figure supplement 1.** Comparison of early tissue opening in the amnion and serosa.

*figure supplement 1*, *Video 3*). We therefore hypothesize that the amniotic rupture competence zone may differ from the rest of the tissue in the nature of its attachment to the serosa (also based on tracking data in *Figure 2—figure supplement 1D–E*), given that the edges of the two tissues then retain apposition throughout subsequent withdrawal (*Figure 3C*: arrow). Before evaluating the progression of withdrawal in detail (below, *Figure 5*), we then completed our assessment of EE rupture by analyzing how preceding events in early development restrict the source of upstream signals to determine the site of rupture.

## The site of rupture is constant in the amnion but not in the serosa

Long-term examination shows that, in contrast to amniotic rupture within specialized cap cells, the relative position of rupture in the serosa is highly variable. At the blastoderm stage, the amnion and

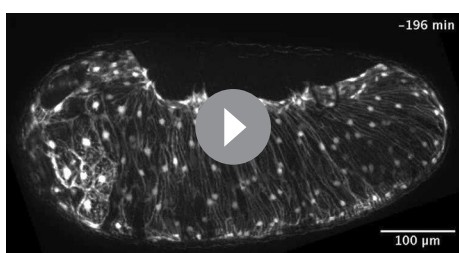

**Video 2.** *Tribolium* extraembryonic tissue rupture and withdrawal shown with amnion-specific EGFP (enhancer trap line HC079). The embryo is shown in lateral aspect with anterior left and dorsal up, with amnion-specific EGFP expression, as well as restricted late embryonic expression domains in the legs and body segments (see also *Figure 1—figure supplement 1*). The movie spans late amnion morphogenesis, from 3.3 hr before rupture through late dorsal closure. Of particular note are the brighter, rounder amniotic cells in an anterior-ventral cap, in which rupture occurs. During withdrawal, the amnion everts (turns inside out), such that the surface that had faced inward toward the embryo is flipped outward to face the vitelline membrane: this is particularly apparent from 35 to 49 min after rupture as the ruptured tissue edge folds over. Time is shown relative to tissue rupture at 0 min. Images are maximum intensity projections with a gamma correction of 0.7 from a z-stack (7 µm step size for a 77 µm stack) recorded every seven minutes over a 9.9-hr time-lapse at 24°C, acquired with an Axio Imager.Z2 with ApoTome.2 structured illumination (Zeiss). Scale bar is 100 µm. *Figure 4C* shows a still at 21 min after rupture.

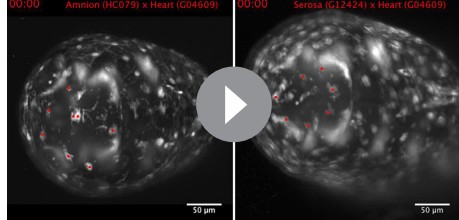

**Video 3.** Extraembryonic rupture filmed at high temporal resolution. Embryos are shown in anterior-ventral aspect with anterior left, and labeled with EGFP from a heterozygote cross of enhancer trap lines labeling the amnion (line HC079) or serosa (line G12424) with selected embryonic domains in the head, segments, and legs (line G04609, 'heart'). During the movie, the EE tissues rupture and withdraw to the extent that the head and first leg pair are exposed. Selected EE nuclei were tracked, where track color segments correspond to fixed units of time (red, orange, yellow, and green: 8.3 min; blue: final 3.3 min). Elapsed time is shown. Images are maximum intensity projections from a z-stack (2.58 µm step size for a 175 or 200 µm stack) recorded every 20 s at 19.5 ± 1°C and shown for a 36.7-minute period (see *Supplementary file 1* for further acquisition details). Single-sided light sheet illumination is from the top, resulting in deterioration of the signal in the lower part of the image due to scattering. The final frame shows time point 36:40 with the approximate position of mouthparts labeled for orientation: An, antennae; Lr, labrum; Mx, maxillae; Lb, labium; T1, first thoracic segment. Scale bar is 50 µm. See also *Figure 4—figure supplement 1* for a visual summary.

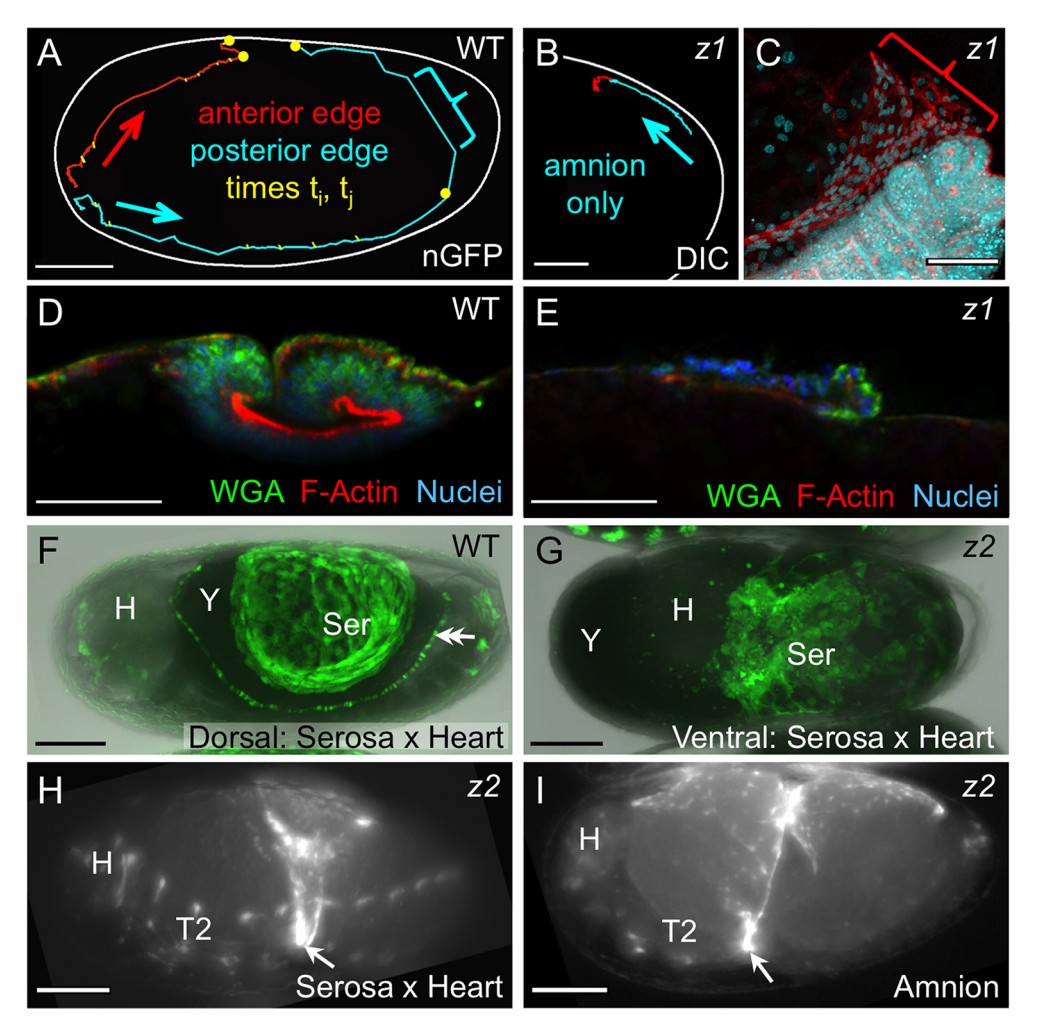

**Figure 5.** Withdrawal dynamics: serosa-driven progression. Images are lateral (A–E,H), dorsal-lateral (F,I) or ventral (G), with anterior left and dorsal up, shown as sagittal optical sections (A–B,D–E) or maximum intensity projections (C,F–I). (A–E) Tracking and histological staining of withdrawing EE tissue edges. The WT EE tissue edge squeezes, then rapidly clears, the abdomen. In A, the blue bracket marks a 1-minute interval within a 6.7-hr total interval for the entire red/anterior and cyan/posterior tracks, at 21°C. The distance of the bracketed track segment from the vitelline membrane (white outline) reflects the degree of squeezing of the abdomen. After *Tc-zen1RNAi* (z1), an amnion-only posterior edge contracts slowly over an uncompressed abdomen. In B, the entire track represents a 3.9-hr interval at 21°C. Note that the track is close to the vitelline membrane outline. Slow amnion-only progression is in part due to ruffling of the tissue. The red track segment in B corresponds to the period of tissue ruffling shown morphologically in the region of the red bracket in C (same staining reagents as in D). Furthermore, there is no apical F-actin enrichment in the folding tissue (E, compare with D and *Figure 3E–F*). (F–G) While WT serosal contraction leads to a dorsally condensed tissue (F), in strong *Tc-zen2RNAi* (z2) phenotypes, the serosa condenses ventrally over the unopened amnion and the confined embryo (G). The double arrowhead labels the cardioblasts; 'Y' marks the opaque yolk. (H–I) In weaker *Tc-zen2RNAi* (z2) phenotypes the EE tissues can rupture and tear ectopically, leaving a 'belt' of EE tissue that squeezes the embryo (arrows). Scale bars are 100 µm (A,F–I) and 50 µm (B–E). Abbreviations as in previous figures.

serosa initially share a lateral tissue boundary within the plane of the blastoderm epithelium (*van der Zee et al., 2005*). As the EE tissues envelop the embryo, they maintain that boundary until they separate into discrete membranes at the serosal window closure stage (*Figure 6A*; *Benton et al., 2013*; *Handel et al., 2000*). As the EE tissues separate, serosal cells rapidly acquire fixed positions under the vitelline membrane (*Koelzer et al., 2014*), while internally the embryo and amnion are not so

tethered. Indeed, we find that over half of all embryos examined (59%, *n* = 69) rotate longitudinally during early germband extension relative to the serosa: up to 90° about the anterior-posterior axis, with a tendency to rest laterally (*Figure 6B*, *Figure 6—figure supplement 1*). More generally, rotation occurs irrespective of whether the embryo's long axis is orthogonal (this study) or parallel (data in *Strobl and Stelzer, 2014*) to gravity, suggesting an inherent anisotropy of the early egg. However, there is no counterpart rotation in later development (*n* = 118), including for 15 embryos filmed continuously and which exhibited typical frequencies of early rotation. In other words, 9 of 15 embryos rotated longitudinally such that the site of serosal window closure (*Figure 6A*: boxed region) did not correspond to the site of later EE tissue rupture (*Figure 6D*: starburst). The time window for rotation ends shortly before the onset of amnion-serosa adhesion to form the bilayer during germband retraction (*Figure 2A*, *Figure 6—figure supplement 1*). Thus, early rotation without any later reversal is a feature of wild type *Tribolium* embryogenesis. This physically uncouples apposition of the mature amnion and serosa from the tissues' relative orientation when they had initially detached from one another.

Given that rupture invariably occurs in the cap region in the amnion, variable longitudinal rotation of the embryo and amnion away from where they had detached from the early serosa therefore precludes an impetus for EE rupture from any eggshell or serosa-specific landmark associated with initial EE tissue separation (contra *Strobl and Stelzer, 2014*). Rather than the serosa causing regionalization in the amnion, it may well be the other way around. We previously observed a subtle morphological change in serosal cells over the yolk compared to those over the amnion, and this change only arises during germband retraction (*Koelzer et al., 2014*), which correlates with the time of EE tissue apposition (*Figure 2A*). Thus, the region of amniotic specialization for rupture may be autonomous (*e.g.*, corresponding to where it ultimately closed at the serosal window stage) or induced by signals from the underlying embryo, but is unlikely to be determined by the serosa.

## Serosal contractility drives withdrawal morphogenesis and demonstrates bilayer adhesion

Once the EE tissues have ruptured anterior-ventrally, they pull back from the embryo as an everting sack that folds up to a dorsal-medial position, similar to a pillowcase being turned inside out as it is peeled off of a pillow (*Figure 6D*). The serosa appears to be the driving force for withdrawal. We had previously observed that the serosa facilitates the final stages of withdrawal during dorsal closure, making the process more robust and efficient (*Panfilio et al., 2013*). Here, we find that this is true throughout these morphogenetic movements, and furthermore that this is an inherent property of this tissue.

Firstly, we again use the serosa-less situation after *Tc-zen1^RNAi^* to assess the serosa's normal contribution. Wild type withdrawal is rapid, with visible squeezing of the embryo's abdomen as the posterior EE tissue edge pulls back and clears the posterior pole. In tracking the edge of the EE tissues during withdrawal, the squeezing is manifest in the degree to which the tissue pulls away from the eggshell (*Figure 5A*: bracketed track segment represents 1 min at 21°C). This is due to serosa-specific enrichment in apical F-actin as the cells undergo a drastic shape change to become pseudocolumnar (*Figure 5D*, also *Figure 3F'*). Indeed, wild type specimens occasionally exhibited a transient extra fold in the posterior EE tissue (*Figure 3C,C'*: double-headed arrow). This likely reflects a degree of stochasticity in tissue relaxation after snapping over the abdomen, similar to kinetic descriptions in other species (e.g., *Patten, 1884*). Both the histological and kinetic situations are altered in the absence of the serosa. In *Tc-zen1^RNAi^* embryos the amnion does withdraw dorsally, but over a much longer time scale and without sufficient force to squeeze the abdomen away from the eggshell (*Figure 5B*: entire track represents 3.9 hr at 21°C). The slow progression is in part due to the tissue ruffling into folds rather than everting via apical constriction (*Figure 5C*: red bracket corresponds to the red track segment in *Figure 5B*). Consistent with a lack of apical constriction, the amnion alone shows no significant F-actin enrichment or increase in cell height (*Figure 5E*).

We then used RNAi against *Tc-zen2*, the functionally diverged paralogue of *Tc-zen1*, as a second approach to functionally test the serosa's role, as strong RNAi knockdown of *Tc-zen2* completely blocks EE rupture (*van der Zee et al., 2005*). In the absence of rupture, the embryo remains confined within the amniotic cavity at the time when dorsal closure is initiated by the epidermal flanks. With nowhere else to go, the flanks grow along the inner surface of the intact amnion until they meet at the ventral midline, resulting in a ventral closure of the body over the legs (*Panfilio, 2008*;

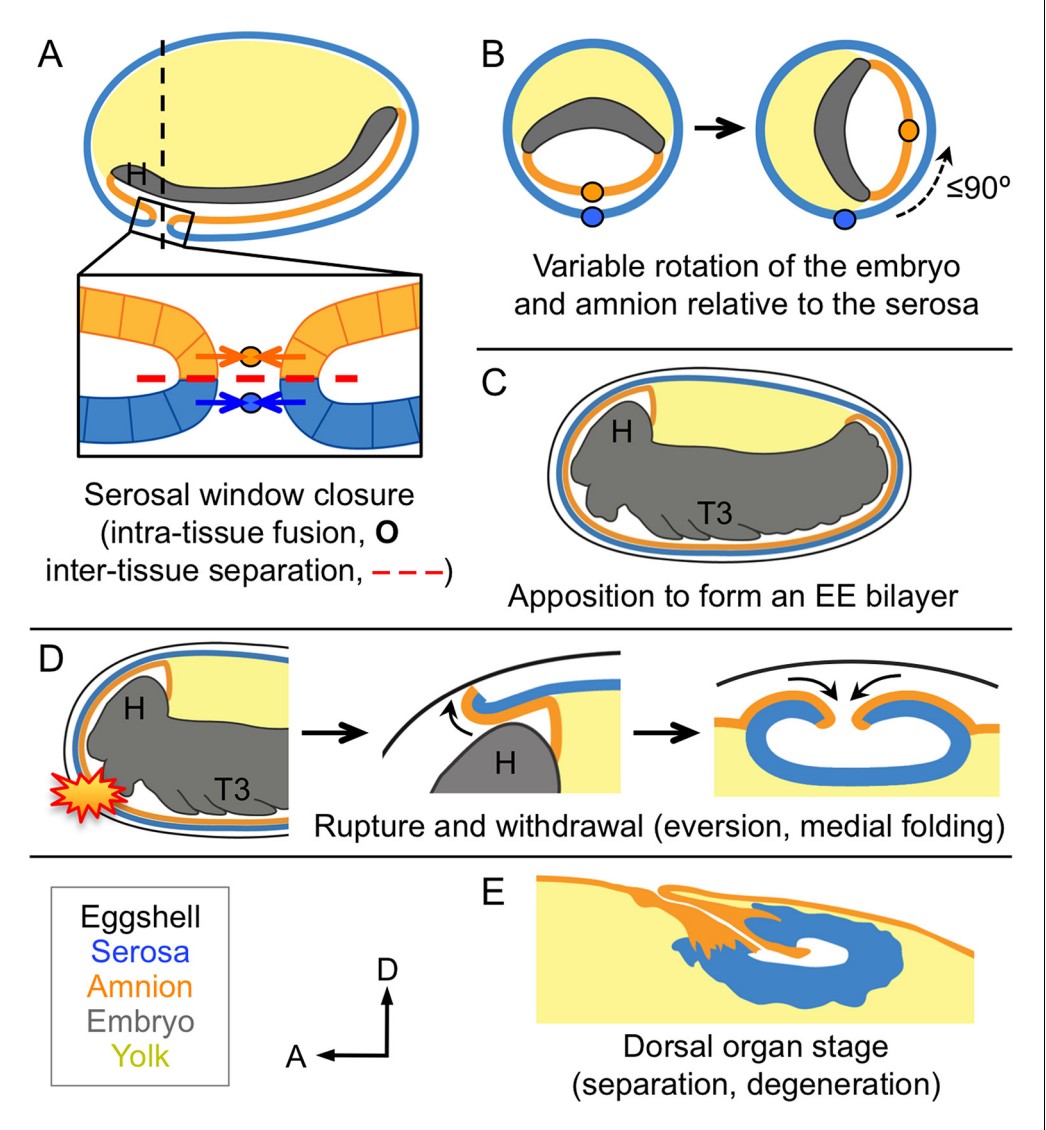

**Figure 6.** Changing amnion-serosa interactions during *Tribolium* extraembryonic morphogenesis. Schematics are sagittal, with anterior left and dorsal up (**A,C–E**), or transverse with dorsal up (**B**), illustrating: initial separation during formation of the amnion and serosa as distinct epithelial covers (**A**), relative rotation in many embryos (**B**: shown at the position of the dashed line in **A**, see also *Figure 6—figure supplement 1*), EE apposition at the retracted germband stage (**C**), the successive stages of EE tissue withdrawal (**D**), and final tissue structure during degeneration at the dorsal organ stage (**E**, see also *Figure 6—figure supplement 2*). Amnion-serosa apposition persists during withdrawal morphogenesis (**D**), which involves EE tissue rupture (starburst), curling over of the resulting tissue edge as it everts (shown for the anterior-dorsal edge), and early serosal compaction as the EE edges fold medially. Abbreviations as in previous figures.

The following figure supplements are available for figure 6:

**Figure supplement 1.** Frequency of embryonic longitudinal rotation during development.

**Figure supplement 2.** Separation during degeneration: the amnion and serosa resolve into two distinct dorsal organ structures.

*Sander, 1976*; *Truckenbrodt, 1979*; *van der Zee et al., 2005*). This manipulation allowed us to examine the serosa's inherent morphogenetic properties without rupture as a trigger event. We find that the *Tc-zen2^{RNAi}* serosa still contracts strongly even in the absence of rupture. Although the amniotic cavity remains unopened and there are no free EE tissue edges, the serosa contracts until it tears ectopically, withdrawing ventrally over the amnion (*Figure 5F–G*: shown mid contraction, when the serosa only occupies a small surface area over the egg). We infer that the serosa remains attached throughout the bilayer region and pulls on the underlying amnion, which would account for the displacement of the embryo's head away from the anterior pole (*Figure 5G*: note yolk anterior to the head). This phenotype and the autonomous nature of serosal contractility in *Tribolium* are conserved compared to *Oncopeltus* (*Panfilio, 2009*). As this contrasts with the differences between these species in amnion structure and behavior, discussed above, it appears that there is a degree of modularity or independence in how the two epithelia have evolved. This is an unexpected conclusion, given the need for tight coordination between these tissues in any one species.

Inter-tissue coordination is further demonstrated in weaker *Tc-zen2^{RNAi}* phenotypes in which rupture does occur but withdrawal morphogenesis is defective. Ectopic tearing of the *Tc-zen2^{RNAi}* EE tissue produces a constrictive EE 'belt' around the embryo. Whether visualized for the serosa (*Figure 5H*) or amnion (*Figure 5I*), the nature of embryonic constriction is consistent with the intact portions of the tissues maintaining adhesion throughout the region of apposition and therefore contracting together. Thus, we propose a model in which correct rupture of the amnion is required for directed EE tissue withdrawal, while the serosa autonomously provides the motive force to achieve this. Adhesion of the EE tissues throughout the region of apposition provides the means to couple these two functions.

## The EE bilayer resolves into two dorsal organs during dorsal closure

In many insects, the dorsal organ is a transient, hollow structure formed by the serosa as it sinks into the yolk and degenerates at the end of its life (*Panfilio, 2008*). Importantly, it has previously been described as comprising only serosal tissue. The amnion is restricted to attachment to the serosa at its lateral edges in many species (*Enslee and Riddiford, 1981*; *Panfilio and Roth, 2010*; *Tojo and Machida, 1997*). Alternatively, it was regarded as fully enveloping the serosa as a smooth outer layer (*Rempel and Church, 1971*), if a distinct amnion was recognized (see first Results and discussion subsection). The persistent bilayered structure of the *Tribolium* EE tissues, however, results in both a serosal dorsal organ and a nested, amniotic dorsal organ (*Figure 6E*, *Figure 6—figure supplement 2*). In light of these observations, re-analysis of late serosa-less *Tc-zen1^{RNAi}* embryos indicates that the previously observed region of anterior-medial amniotic F-actin enrichment (*Panfilio et al., 2013*) corresponds to the amniotic dorsal organ, demonstrating that this structure also forms in the absence of a serosa. Thus, at the end of the EE tissues' lifetimes, they once again function as independent structures, involving separation of the bilayer as the tissues degenerate separately (*Figure 6—figure supplement 2*).

## Conclusions

Altogether, the tissue reorganizations for insect EE withdrawal have complex implications for inter-epithelial attachment, balancing a requirement for tissue continuity over the yolk and coordinated withdrawal with enabling the amnion and the serosa to follow their own morphogenetic programs. It was previously known that the *Tribolium* EE tissues arise from the same blastodermal cell sheet before enclosing the embryo and detaching from one another (*Figure 6A*; *Benton et al., 2013*; *Handel et al., 2000*; *Koelzer et al., 2014*). After those early stages, the structure and arrangement of the amnion had been obscure. Here, we have characterized a new genetic resource in the form of the HC079 enhancer trap line that literally illuminates this enigmatic tissue for the first time. Following its early detachment from the serosa, the amnion often rotates relative to the serosa (*Figure 6B*) before reattaching to form the bilayer (*Figure 6C*). Then, given that the EE tissues withdraw from the embryo as a single unit (*Figure 6D*), the re-establishment of their independence during final degeneration at the dorsal organ stage is striking (*Figure 6E*).

The mechanical requirements of these morphogenetic events imply a precisely regulated and dynamic mode of epithelial attachment. The apposed surfaces (*Figure 6C–D*) present a basal-basal interface where they presumably interact via the basement membrane, which is known to influence

both cell adhesion and shape changes at tissue folds during embryogenesis in many animal systems (*Daley and Yamada, 2013*). Furthermore, the bilayer structure and its rupture in *Tribolium* have intriguing parallels with vulval development in *C. elegans*, where two apposed epithelial sheets must also be opened. In the roundworm, the uterine and ventral epidermal layers are juxtaposed but separated by their respective basement membranes (*Morrissey et al., 2014*). This barrier is breached when the specialized anchor cell of the inner uterine tissue initiates a sequence of extracellular remodeling events to connect the two layers (*Hagedorn and Sherwood, 2011*). It will be interesting to see whether similar approaches to local basement membrane removal and gap widening (*Ihara et al., 2011*) also occur during insect EE rupture, a rapid event that occurs nearly an order of magnitude more quickly than anchor cell invasion (*Figure 4F*; *Hagedorn and Sherwood, 2011*). Also, in contrast to a system of invasive cell behavior in which cell cycle regulation and the balance of mitosis among neighboring cells contribute to basement membrane breach and widening, respectively (*Matus et al., 2014*; *2015*), the insect EE tissues enter the endocycle and mature to a polyploid state before rupture (*Panfilio et al., 2013*). The withdrawing EE tissues of *Tribolium* thus provide a new system for investigating restructuring of the basement membrane and the cellular-extracellular matrix in simple epithelia.

Our working model for how the amnion and serosa cooperate to achieve withdrawal also generates several testable hypotheses regarding the nature of their morphogenetic interactions. For example, we observe a difference in rupture initiation between the amnion and serosa (*Figure 4F–G*, *Video 3*) that we interpret as a delay before the serosa ruptures and then catches up relative to the amnion. Curiously, serosal cells seem to flatten (lose peripheral EGFP signal) prior to rupture (compare *Figure 4E1-E2*), and partial dissociation of some serosal cells (*Figure 4E2*: blue cells) is observed during rupture, although epithelial integrity is maintained during withdrawal. Might this represent a more brittle approach to rupture in which the serosa effectively shatters, compared to the flexibility of opening amniotic cells? In future work, it will be important to elucidate how cells in the amnion and serosa differ in their mechanical properties (*Koehl, 1990*). In *Drosophila* it was recently shown that non-muscle myosin II plays a nuanced role in tuning the mechanical properties of the peripodial epithelium so that it can rupture and withdraw from the wing imaginal disc columnar tissue (*Aldaz et al., 2013*). The future development of comparable genetic tools in *Tribolium* will permit the direct testing of our model.

Taking a step back from rupture itself, it will also be important to uncover the upstream genetic determinants for this event. For example, *Tc-zen2* not only promotes rupture but is also expressed in the anterior amnion in early development (*van der Zee et al., 2005*), which approximately prefigures the region in which we later detect the morphologically distinct amniotic cap cells. However, this transcription factor is also expressed throughout the serosa, and we find that its knockdown impairs both EE tissues without specifically affecting amnion anterior-ventral differentiation (data not shown), indicating that other regulators control cap formation. Identifying the molecular cues for amniotic regionalization and the proximate triggers for rupture itself will further clarify the amnion's role.

Meanwhile, the bilayered EE arrangement in *Tribolium* is thus far unique among insects and resolves a long-standing ambiguity regarding amnion structure and function. While limited data are available for holometabolous insects with complete EE tissues, hemimetabolous insects have been more extensively studied and have a different arrangement. Hemimetabolous EE withdrawal, known as katatrepsis, involves the embryo being pulled out of the yolk by the EE tissues. The requirements for katatrepsis restrict amnion-serosa connection to a distinct border region with lateral-lateral cell contact (*Panfilio, 2009*; *Panfilio and Roth, 2010*), but katatrepsis was lost at the base of the holometabolous insect radiation (*Panfilio, 2008*). The longitudinal rotation of the young *Tribolium* germband embryo and amnion is akin to the more extensive movements of these tissues within the yolk in hemimetabolous insects (*Cobben, 1968*; *Rakshpal, 1962*; *Wheeler, 1889*). In contrast, the basal rather than lateral nature of reattachment in *Tribolium* represents a qualitatively different physical environment for the amnion during degeneration (*Figure 6E*, *Figure 6—figure supplement 2*; *Horn et al., 2015*). Consistent with this topographical change, we observe that whereas the morphogenetic properties of the serosa are conserved between *Tribolium* and the hemimetabolous bug *Oncopeltus*, the structure and mode of preparation for rupture in the amnion in these species are decidedly different. Further taxonomic sampling will reveal whether *Tribolium* represents the norm or one of multiple possible EE configurations within the Holometabola. Loss of katatrepsis in this

insect lineage may have relaxed constraints on extraembryonic developmental strategies and permitted a degree of independence in the evolution of the insect amnion and serosa.

## Materials and methods

### *Tribolium* stocks and RNAi

Analyses were performed in the San Bernardino wild type strain, EFA-nuclear-GFP (nGFP) line (*Sarrazin et al., 2012*), and selected GEKU screen enhancer trap lines (*Trauner et al., 2009*): G12424 ('serosa') and G04609 ('heart': cardioblasts and segmental domains), as described (*Koelzer et al., 2014*); and HC079 ('amnion', characterized in this study, see also *Figure 1—figure supplement 1*, *Video 1*). The KT650 serosal line (*Koelzer et al., 2014*) was also used in assessing pre-rupture serosal cell morphology throughout the region of apposition with the amnion (as in *Figure 4A*). Parental RNA interference (RNAi) for *Tc-zen1* and *Tc-zen2* was performed as previously described, with adult females injected with double-stranded RNA of ≥688 bp at 1 µg/µl concentration (*Panfilio et al., 2013*; *van der Zee et al., 2005*).

### Live imaging and rendering

Embryos were dechorionated and mounted on slides in halocarbon oil for conventional microscopy (*Panfilio et al., 2013*). Additionally, a light sheet fluorescence microscope (mDSLM model: *Strobl and Stelzer, 2014*) was used for high temporal and spatial resolution recordings. Here, embryos were dechorionated and embedded in low melt agarose and filmed in anterior-ventral aspect (see *Supplementary file 1* for acquisition details for data in *Figure 4D,E,G*, *Figure 2—figure supplement 1D–E*, and *Video 3*). Temporal resolution was improved both by image acquisition speed of this microscope system, and, for certain applications, by slowing the rate of embryogenesis via temperature regulation. Whereas embryogenesis takes approximately three days at the customary stock maintenance temperature of 30°C (*Koelzer et al., 2014*), selected embryos were filmed for up to 24 hr at 19.5 ± 1°C, slowing development roughly by a factor of four (*Sokoloff, 1974*). Subsequent hatching was confirmed in all cases to verify that recordings documented healthy development. With this criterion, we analyzed rupture with light sheet fluorescence microscopy in 14 embryos, across homozygous and heterozygous genetic backgrounds. In fact, of the 17 embryos observed, only one died prior to hatching, while in two cases we were unable to confirm hatching due to post-acquisition handling difficulties.

Cell tracking was performed on maximum intensity projection (MIP) time-lapse movies with MTrackJ (*Meijering et al., 2012*). Cell outlines were drawn manually in ImageJ with the segmented line and polygon selection tools, based on observed cell morphology.

### Rupture initiation analysis

Embryos were recorded every 3.5 min at 27.5–28°C on a DeltaVision RT microscope (Applied Precision/GE Healthcare Life Sciences, Issaquah, Washington, USA), with simultaneous recording of up to 11 embryos each of the serosa-heart and amnion-heart crosses in each of three experiments. Using the MIP movie output, rupture initiation was determined as the interval between the following events. The start of rupture was recorded as either the first frame in which two or more cells clearly diverged, followed by the appearance of a hole in the EE tissue at the same position in subsequent frames, or as the first frame after a sudden collapse of the anterior EE membrane, whichever came first. The end of rupture was recorded as the first frame in which the retracting membrane cleared the head dorsally.

### Longitudinal rotation analysis

Embryos were analyzed from time-lapse data sets of at least 15.75 hr duration at 30°C. Orientation and degree of rotation were determined by eye from MIP movies, and categorized into eight sectors of 45° around the egg circumference. See *Figure 6—figure supplement 1* for further details.

### Histological staining

To assess early amnion topography, fuchsin staining after standard fixation and methanol shock without complete devitellinization was performed based on standard protocols (*Wigand et al., 1998*).

For fluorescent imaging of endogenous EGFP, embryos were fixed, manually devitellinized, optionally stained with fluorescently conjugated phalloidin or wheat germ agglutinin (WGA), and embedded in Vectashield mountant with DAPI as described previously (*Panfilio et al., 2013*).

## Acknowledgements

We thank Sue Brown for providing many enhancer trap lines; Gianluca Sharbaf Azari, Max Kornilov, and Max Pentzien for contributions during student research; Frederic Strobl, Sven Plath, Ernst Stelzer, and Pavel Tomancak for training and advice on light sheet microscopy; Ferdinand Grawe (Microscopical Anatomy and Molecular Cell Biology Research Group, Institute for Anatomy I., Uniklinik Köln) for assistance with electron microscopy; Siegfried Roth for discussions and feedback on the manuscript. We also thank the editor and our peer reviewers for their dedication in improving the manuscript, including Dave Matus for expanding our horizons and Urs Schmidt-Ott for ongoing discussions on insect EE development and evolution. In this latter context, Urs kindly shared a presubmission draft of a new review on this topic (*Schmidt-Ott and Kwan, 2016*).

This work was supported by the German Research Foundation (Deutsche Forschungsgemeinschaft) Emmy Noether Program grant number PA 2044/1-1 to KAP. Additional support for the mDSLM light sheet microscope was provided by the former Collaborative Research Center 572 ("Commitment of Cell Arrays and Cell Type Specification") and the currently running Collaborative Research Center 680 ("Molecular Basis of Evolutionary Innovations"), both funded by the German Research Foundation.

## Additional information

### Funding

| Funder | Grant reference number | Author |
| --- | --- | --- |
| Deutsche Forschungsgemeinschaft | Emmy Noether Program, PA 2044/1-1 | Kristen A Panfilio |

The funders had no role in study design, data collection and interpretation, or the decision to submit the work for publication.

### Author contributions

MH, Designed the rupture initiation experiments, Designed the light microscopy experiments, Conducted the experiments, Conception and design, Acquisition of data, Analysis and interpretation of data, Drafting or revising the article; TH, Designed the rupture initiation experiments, Conducted the experiments, Conception and design, Acquisition of data, Analysis and interpretation of data, Drafting or revising the article; SK, Planned the RNAi and fuchsin experiments, Conducted the experiments, Acquisition of data, Analysis and interpretation of data, Drafting or revising the article; KAP, Conceived the experiments, Planned the RNAi and fuchsin experiments, Designed the rupture initiation experiments, Conducted the experiments, Conception and design, Acquisition of data, Analysis and interpretation of data, Drafting or revising the article

### Author ORCIDs

Kristen A Panfilio, http://orcid.org/0000-0002-6417-251X

## Additional files

### Supplementary files

• Supplementary file 1. Acquisition parameters for featured mDSLM light sheet experiments.

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
