## [Decision Letter]

[Editors’ note: a previous version of this study was rejected after peer review, but the authors submitted a revised version for reconsideration.]

Thank you for submitting your work entitled "The beetle amnion and serosa functionally interact as apposed epithelia" for further consideration at *eLife*. Your article has been favorably evaluated by Diethard Tautz (Senior editor), Marianne Bronner (Reviewing editor) and two new reviewers, one of whom, David Matus, has agreed to share his identity.

The manuscript has been improved but there are some remaining issues that need to be addressed. As you will see below, Reviewer 1 is very positive and thinks the manuscript is appropriate for *eLife* with some relatively minor revisions. Reviewer 2 thought the manuscript was more appropriate for a specialized journal. On balance, I find the paper potentially acceptable for publication pending the minor revisions suggested by the reviewers. Please see the comments below for details.

Reviewer #1:

Hilbrant et al. provide the first live-cell imaging study to understand how insect extraembryonic (EE) tissues (amnion and serosa) rupture and withdraw during the embryogenesis of the holometabolous beetle *Tribolium castaneum*. Novel insights are gained through the use of two different GFP transgenic lines, labeling the amnion and serosa, independently. As a previous reviewer has suggested, having a dual labeled (green/red) line would of course greatly aid their studies, but I understand as a non-canonical model system that the creation of this transgenic line is beyond the scope of this study, and the authors have done their due diligence using the two independent strains to the best of their ability, using a combination of structured illumination and light sheet imaging to resolve these morphogenetic events for the first time in live embryos. Thus, methodologically this represents a significant advance for their field. In terms of insights made in understanding the process at a cell biological level, the authors find striking differences between cell movements during rupturing in the two tissues, in timing, localization of the initial rupturing event, cell morphologies during rupturing, and coordination of the event itself. I am particularly intrigued by the stochastic nature of the event in the amnion, as the authors reported multiple cases of intracellular openings, as shown in Figure 4. This was not observed in the serosa. Additionally, the serosa differs in its morphogenetic behavior, as rupture initiation seems more random. The authors use knockdown of two *zen* paralogs to demonstrate the functional importance of the serosa, as depletion of *Tc-zen1* by RNAi results in loss of enriched apical F-actin and a slowing of amnion dorsal withdrawal.

In their Conclusion, the authors hypothesize that regulation of adhesion to the basement membrane may be critical to these morphogenetic movements, although currently they lack the tools to visualize the basement membrane. I think it will be critical to be able to visualize the basement membrane and functionally examine adhesion mechanisms in both epithelia going forward, but do not think those experiments fall within the scope of the current manuscript either.

I was not involved in the initial review of this manuscript, but appreciated the authors' response to the reviewers' criticism. In my opinion the authors have addressed the previous reviewers’ concerns adequately, especially through newly acquired data with light-sheet imaging. Particularly, the new data presented in Figure 4 are compelling, and open up a fascinating new area of study (inter vs. intracellular rupturing in epithelia during morphogenesis). I am unaware of any other system where one might be able to examine this with live-cell imaging approaches, and think the inclusion of these data and the discussion in this version of the manuscript should make the paper of interest to a broader audience beyond the insect evo-devo and morphogenesis community, though I have a few recommendations of other papers that might be worth mentioning in the Discussion (see below). In summary, I have no substantive concerns about the manuscript in its current form.

*Reviewer #1 (Minor Comments):*

1) Could you add "transgenic" into the Abstract here: "Here we use new fluorescent transgenic lines in the beetle…".

2) I am not sure if *eLife* has rules concerning figure referencing, but I found it somewhat confusing that figures were referenced out of order throughout the manuscript (for first time they were mentioned). I understand that figure layout often fits together in specific ways, but it would have made it easier to go through the manuscript if the figures were referenced in order whenever possible.

3) During *C. elegans* uterine-vulval attachment there are two basement membranes that are apposed to one another that initially slide back and forth. Following anchor cell invasion, the basement membranes become fused, through the secretion of the extracellular matrix molecule hemicentin (see Morrissey et al. Dev Cell 2014).

4) It might be worth discussing Srivastava et al. PNAS 2007 (for Drosophila) and any of the work by Guojun Sheng's group (e.g., Nakaya et al. NCB 2008, Nakaya et al. JCB 2013) and several specific papers from the Sherwood group (Ihara et al. NCB 2010; Matus et al Nat Comm. 2014), rather than citing the review of Hagedorn and Sherwood 2011, as other systems that have looked at hole formation and basement membrane remodeling in vivo (it's a very small literature!).

Reviewer #2:

Hilbrant et al. report on the rupture and withdrawal of the amnion and serosa in *Tribolium*. They use advanced microscopy techniques and generate new tissue markers, which is no small achievement in *Tribolium*. The imaging is generally of high quality. However, the study is largely descriptive (albeit very carefully descriptive and examined to the level of individual cell morphologies) and, in my opinion, some kind of mechanical or cell biological interventions would be required to fully test their hypotheses and produce a body of work suitable for *eLife*. Additionally, the absence of two-color imaging (and I appreciate that this is highly difficult with the current *Tribolium* technologies) makes it difficult to be 100% certain of some of the relationships reported between the amnion and serosa and how they evolve temporally. The work does provide important insights into *Tribolium* development, but it seems more appropriate for a specialty development journal.

*Reviewer #2 (Minor Comments):*

The statement, “However, we find that [*Tc-zen2*] knockdown impairs both EE tissues without specifically affecting amnion anterior-ventral differentiation, indicating that other regulators control cap formation. In future work, identifying the molecular cues for amniotic regionalization and the proximal triggers for rupture itself will further clarify the amnion’s role” is more appropriate for the Conclusion, and is distracting in the context of a Results section.

Some subdivision of the larger Results sections would be nice. For example, there are seven paragraphs in the "The amnion initiates EE withdrawal" subsection. Many different ideas and interpretations are reported here, and having subheadings to describe each of these would make the writing more user-friendly to the casual reader.

---

## [Author Response]

Reviewer #1 (Minor Comments):

*1) Could you add "transgenic" into the Abstract here: "Here we use new fluorescent transgenic lines in the beetle…".*

We have added the term and agree with the reviewer that this improves precision.

*2) I am not sure if eLife has rules concerning figure referencing, but I found it somewhat confusing that figures were referenced out of order throughout the manuscript (for first time they were mentioned). I understand that figure layout often fits together in specific ways, but it would have made it easier to go through the manuscript if the figures were referenced in order whenever possible.*

We acknowledge that finding a strictly linear order to the presentation of the material was tricky, and we have now done our best to improve this. All figures are cited in the main text in sequential order, with a “road sign” comment to situate Figure 5 relative to the other results content. To avoid early mention of the summary illustrations in Figure 6, we have now added thumbnail cartoons to Figure 2 and Figure 3 so that the relevant anatomical features are shown and referenced locally (please see Figure 2, Figure 3, the second paragraph of the Results section and the corresponding figure legends). While the illustrations in Figure 6 support multiple sections of the results, we felt that the presentation worked best by keeping these original images together in a single, concluding figure.

Also, please note that in addition to these tweaks to Figure 2 and Figure 3, we have uploaded a corrected version of Figure 2—figure supplement 1. In panel D4, the previously uploaded version inadvertently showed the serosa at 2 minutes after rupture, rather than time 0, and the correct still image is now shown.

*3) During C. elegans uterine-vulval attachment there are two basement membranes that are apposed to one another that initially slide back and forth. Following anchor cell invasion, the basement membranes become fused, through the secretion of the extracellular matrix molecule hemicentin (see Morrissey et al. Dev Cell 2014).*

We thank the reviewer for pointing out this error, and we have corrected the text accordingly.

*4) It might be worth discussing Srivastava et al. PNAS 2007 (for Drosophila) and any of the work by Guojun Sheng's group (e.g., Nakaya et al. NCB 2008, Nakaya et al. JCB 2013) and several specific papers from the Sherwood group (Ihara et al. NCB 2010; Matus et al Nat Comm. 2014), rather than citing the review of Hagedorn and Sherwood 2011, as other systems that have looked at hole formation and basement membrane remodeling in vivo (it's a very small literature!).*

We further thank the reviewer for the suggested reading list, which indeed helped us to better approach this body of literature and to find additional cell biological features to compare with our own system. The text has been updated accordingly.

Reviewer #2 (Minor Comments):

*The statement, “However, we find that [Tc-zen2] knockdown impairs both EE tissues without specifically affecting amnion anterior-ventral differentiation, indicating that other regulators control cap formation. In future work, identifying the molecular cues for amniotic regionalization and the proximal triggers for rupture itself will further clarify the amnion’s role” is more appropriate for the Conclusion, and is distracting in the context of a Results section.*

We have now moved this material on *Tc-zen2* and the genetic determination of the rupture site to its own paragraph in the Conclusions section, and we ensured that the *Tc-zen2* gene is still adequately introduced at the first mention.

*Some subdivision of the larger Results sections would be nice. For example, there are seven paragraphs in the "The amnion initiates EE withdrawal" subsection. Many different ideas and interpretations are reported here, and having subheadings to describe each of these would make the writing more user-friendly to the casual reader.*

We have amended the structure of the Results and Discussion sections in line with this helpful suggestion. The previous, large “The amnion initiates EE withdrawal” section has been subdivided into three sections, and we have updated the heading “Serosal contractility drives withdrawal morphogenesis [and demonstrates bilayer adhesion]” (new text indicated by brackets) to more clearly indicate the content here. Transitional sentences and phrases have also been interspersed to help guide the reader through this material.